# Priorities for developing stroke care in Ireland from the perspectives of stroke survivors, family carers and professionals involved in stroke care: A mixed methods study

Eithne Sexton[1]*, Karen Fowler[1], Anne Hickey[1], David J. Williams[2,3], Frances Horgan[4], Elaine Byrne[5], Chris Macey[6], Padraic Cuffe[6,7], Suzanne Timmons[8], Kathleen Bennett[1]

1 School of Population Health, RCSI University of Medicine and Health Sciences, Dublin, Ireland, 2 Department of Geriatric and Stroke Medicine, RCSI University of Medicine and Health Sciences, Dublin, Ireland, 3 Department of Geriatric and Stroke Medicine, Beaumont Hospital, Dublin, Ireland, 4 School of Physiotherapy, RCSI University of Medicine and Health Sciences, Dublin, Ireland, 5 Centre for Positive Health Science, RCSI University of Medicine and Health Sciences, Dublin, Ireland, 6 Irish Heart Foundation, Dublin, Ireland, 7 Patient Collaborator, Sligo, Ireland, 8 Centre for Gerontology and Rehabilitation, University College Cork, Cork, Ireland

* eithnesexton@rcsi.ie

**Data Availability Statement:** Raw interview transcripts cannot be shared to ensure anonymity of participants. However, detailed aggregate data is

## Abstract

### Introduction

Increasing numbers of people are living with stroke, due to population ageing and improved survival, leading to a need for evidence to inform future policy decision-making. This study aimed to engage with stakeholders in Ireland to identify priorities for stroke services development.

### Methods

A sequential mixed methods design was used. Phase 1 (qualitative) was exploratory, involving initial priority gathering via an online qualitative survey and interviews, with stroke survivors, family/main carers, and professionals working in stroke care. Framework analysis was used to generate a long-list of improvements to stroke services. Phase 2 involved a quantitative survey, where stakeholders selected five priority improvements from the long-list. Results were discussed in a stakeholder meeting.

### Results

In-depth interviews were completed with 18 survivors, 13 carers and 8 professionals, while 80 professionals took part in a qualitative survey (phase 1). Priority areas of care were identified and a long-list of 45 priority improvements was generated. In phase 2, 34 survivors, 19 family carers and 42 professionals completed a survey. The highest priority improvements (selected by >20% of respondents) were access to specialist neuro-rehabilitation, ongoing support for life after stroke, recruitment/retention of specialist staff, improved information and support for health system navigation, and access to specialist acute care. Stroke

available from the Open Science Repository: https://osf.io/3dvfs/.

**Funding:** ES received funding to carry out this research from the Health Research Board Ireland [ARPP-2020-010].

**Competing interests:** The authors have declared that no competing interests exist.

survivors/carers prioritised exploring ways to improve access for strokes with atypical presentation, while professionals prioritised specialist inpatient rehabilitation and early supported discharge. Neither group prioritised stroke prevention. Based on discussions in the stakeholder meeting (n = 12), it was decided that support for mental health should also be included as a priority.

## Discussion

The development of stroke services benefits from exploring the priorities of those receiving and delivering stroke care. Findings emphasise the need for equitable access to high quality adequately-staffed services, particularly post-discharge, that are easy to navigate, with good communication, and effective information provision.

## Introduction

Stroke is a major cause of death and disability [1]. In 2021, 10% of stroke patients admitted to hospital in Ireland died while in hospital, 34% had moderate or severe disability at discharge, and 31% of patients had mild disability [2]. Approximately 20% of stroke survivors have dementia [3], with a further 40% having some level of cognitive impairment that does not meet the full criteria for dementia [4]. Improved survival post-stroke and changing population age structures are leading to rapid increases in the number of stroke survivors. Epidemiological modelling of stroke prevalence in people aged 40–89 in Ireland, using the WHO DISMOD approach, indicated a total prevalence of 49,140 (23.1 per 1000 pop) in 2016, projected to increase to 75,807 (24.3 per 1000 pop) by 2035 [5]. Stroke services need to meet this increasing demand for post-stroke rehabilitation and community services, whilst also keeping pace with ongoing rapid advances in acute management. There is thus an urgent need to develop and improve services across the stroke care continuum, and to incorporate the views of stakeholders in this service development process.

Recent audit of the 24 hospital-based stroke services in Ireland found that all have a stroke unit, almost all (22/24) provide a 24/7 thrombolysis service, and half provide Early Supported Discharge (ESD) (a home-based rehabilitation service) [6]. Two centres provide a 24/7 thrombectomy service. However, the number of stroke unit beds is insufficient to meet demand, and staffing is significantly below recommended levels for nursing and rehabilitation staff, particularly psychologists. Similar audit has not been carried out on non-hospital services. There is a policy to develop a national managed clinical rehabilitation network [7], but implementation has been slow, partly due to low political priority [8].

There is a strengthening recognition of the need to engage healthcare users in informing service development and improving quality of care [9]. Stroke clinical guidelines recommend that quality improvement should be informed by patient perspectives and experiences [10]. Previous qualitative and mixed methods studies of stakeholder views and experiences in relation to stroke services have found particular challenges related to continuity/co-ordination of care [11–13], information provision [11–13], psychological support [12], and inequitable access to care, particularly based on geographical variation in services [13]. In Ireland, a previous national survey of stroke survivors found that emotional problems and fatigue had the highest levels of unmet need [14].

The aim of this study was to engage with stakeholders in Ireland to identify their priorities for the development of stroke prevention and management strategies and policies for Ireland.

This builds on previous survey-based research on post-stroke needs in Ireland [14] by using a rigorous, sequential mixed methods approach to identify not just gaps in service provision, but priorities for improving stroke care, including both qualitative data on a broad set of priorities, and more specific quantitative data on which improvements are considered most important. Stakeholders included 1) stroke survivors; 2) family member/main carers; and 3) professionals delivering or otherwise involved in stroke care (including healthcare professionals, and staff working in research, policy or advocacy).

An additional aim of the study was to identify a set of five priority improvements, that would be further evaluated using population-based modelling and economic evaluation, using a previously developed model [5]. Population-based modelling is a key tool to support decision making in relation to further development of stroke services, and can benefit from being informed by stakeholder priorities and concerns.

## Methods

A protocol for this study has been published [15]. The Consolidated Criteria for Reporting Qualitative Research (COREQ) [16] a 32-item checklist was used and reported (see S1 Table). Further supplementary material is available on Open Science Framework (OSF): https://osf.io/3dvfs/?view_only=c46dbae2cde44a55be8c1e0fb3980c36, DOI 10.17605/OSF.IO/3DVFS. The study was granted ethical approval from the RCSI Research Ethics Committee (REC) (refs 202101017 and 202111016). A steering group for the study included leaders working in stroke research, care delivery, advocacy and policy, and a stroke survivor.

### Study design

The research design was an exploratory sequential mixed methods study, based on the approach outlined by the James Lind Alliance (JLA) for identifying research priorities [17], re-purposed to identify priorities for improving services. Phase 1 involved a qualitative approach for initial priority gathering, based on an open-ended (qualitative) online survey and interviews. Framework analysis [18] was used to identify a long-list of specific priority improvements to be used in a second phase of data collection. Phase 2 involved a quantitative survey, where participants were asked to rank their five priority improvements from the initial long-list. The phase 2 survey results were also used to inform a final stakeholder meeting, where a small group of stakeholders (n = 12) decided on a final set of priority improvements, to be evaluated in further detail using epidemiological and economic modelling.

### Participants

Three groups of stakeholders were invited to participate across phases 1 and 2 of the study– 1) stroke survivors, 2) family or main carers of stroke survivors, and 3) professionals involved in stroke delivery in Ireland, including healthcare professionals (HCPs) and people working in research, policy and advocacy, and senior clinical and policy leaders.

### Phase 1 recruitment

Purposive sampling was used. Stroke survivors and family member/main carers were recruited for interviews through multiple sources. This included social media platforms and a range of voluntary organisations representing and providing services to stroke survivors in different parts of the country, including those catering to specific groups such as survivors with aphasia. This helped to ensure a heterogeneous group that was as reflective as possible of the stroke population. Participants were not known to the interviewer prior to the study. Detailed

information on participant recruitment and inclusion criteria is provided in the protocol [15]. Briefly, inclusion criteria for stroke survivor/main carer participants were: 1) had (or cared for someone who had) a diagnosis of stroke more than 6 months prior; 2) aged 18 years and over and resident in the Republic of Ireland; 3) able to converse in and understand English; and 4) capable of providing informed consent. Sample consent forms and information leaflets are available at the OSF link above, including aphasia-friendly materials and accessible materials for individual with cognitive impairment. The recruitment target was 12–15 stroke survivors and 12–15 carers. We also aimed to carry out a small number of interviews (8–10) with individuals identified as key clinical, policy and research leaders in stroke.

Recruitment for the interviews began in April 2021. As recruitment progressed, we identified gaps (e.g., geographical area, severity, specific patient/service experiences) and re-targeted recruitment strategies accordingly. For example, we expanded the criteria to include people who had, or were caring for someone with, a diagnosis of vascular dementia (as a result of stroke or other vascular pathology, such as multiple transient ischemic attacks). Recruitment continued until March 2022, with data collection concluding in April 2022. The decision to stop recruitment was made in consultation with the study steering group, on the basis of data saturation along with feasibility and time constraints. Data saturation was considered at two levels–code saturation (no additional issues were identified) but also meaning saturation (no further dimensions, nuances or insights of issues were identified) [19].

HCPs and policy/research/advocacy stakeholders were recruited for the online qualitative survey through multiple approaches (see protocol for more detail [15]). Information about the study was disseminated through social media, personal networks of the research team and collaborators, a range of professional organisations and networks, and through voluntary bodies representing and providing services to stroke survivors in Ireland. Professionals had to be involved in stroke care delivery in the Republic of Ireland, either directly or in a policy/research/advocacy role. Survivors and carers were given the option of completing the online survey if they preferred that to an interview. Recruitment for the survey began in July 2021 and was open until November 2021. The target for recruitment was a minimum of 80–100 responses.

The interviews were primarily targeted at survivors and carers, although they were also given the option of completing the online survey if they wished. It was anticipated that the interviews would be easier for some individuals in this group, due to varying levels of computer literacy and access to technology. More importantly, the phase 1 survey questions were open-ended, requiring potentially lengthy written responses, and more effort than a verbal interview. Consistent with this, few survivors and carers availed of the survey option (n = 8). The interview format allowed us to generate highly relevant rich narrative data from survivors and caregivers on their experiences.

## Phase 2 recruitment

Phase 1 participants who consented to be contacted again were invited to participate in a Phase 2 survey, either by email or text message. We also recruited further participants through the same channels as Phase 1 –social media, voluntary organisations, and professional bodies. The inclusion criteria were the same as for Phase 1. Recruitment for the survey took place from November 2022 to January 2023. As participation in the Phase 2 survey was through an anonymous link, it was not possible to determine how many of these participants also took part in Phase 1.

We decided that a target sample size of 96 would be feasible, with a 10% margin of error in estimates of the proportion of participants choosing each improvement, when considering the

total sample [20]. A minimum of 43 participants in the professional group, and 43 in the combined survivor and carer group, would allow estimates specific to these groups with a 15% margin of error. We considered that this was a reasonable level of precision, as the survey results were not the final stage of the process, but were discussed and finalised in a stakeholder meeting, as per the JLA process.

## Stakeholder meeting

In February-March 2023, a small group of stakeholders (target n = 12, balanced across stakeholder groups) were invited to take part in a meeting, based on the nominal group technique [19], to decide on a set of priorities to be examined further using epidemiological and economic modelling, using an already established population-based model [5]. Participants were randomly selected from the list of Phase 2 survey participants who indicated an interest in attending a meeting.

## Informed consent

All participants at each stage provided written informed consent. For the interviews, consent was assessed on a continual basis, following the principle of process consent–at the initial consent form stage, immediately prior and during the interview, and after the interview has concluded. It was assumed that individuals had the capacity to consent, unless otherwise demonstrated, as per the Assisted Decision-Making Capacity Act (2015) and the National Consent Policy [21]. The consent process and all aspects of data management are described in detail in the study protocol [15].

## Data collection and analysis

**Phase 1: Initial priority gathering.**   Qualitative interviews and an open-ended online survey were conducted in parallel. Semi-structured topic guides were developed for the interviews, tailored to survivors, carers and senior leaders. The online survey included nine open-ended questions, covering different areas of stroke care, and tailored to each participant group. Both topic guides and survey questions were informed by the relevant literature [11, 22–27]. The survey was piloted with 4 participants (1 survivor, 2 HCPs and 1 researcher), resulting in minor revisions. All materials were reviewed by the study steering group. Sample copies of data collection materials are available at the OSF link above. The development and content of the data collection materials is described in more detail in the study protocol [15].

Interviews were conducted by the lead researcher ES, a female experienced post-doctoral health services researcher with training in qualitative data collection and analysis. Interviews were audio recorded, and field notes recorded before and after the interview. The topic guide was revised iteratively as data collection progressed. Participants were given the option of reviewing the transcript, with all senior leaders and 16/29 survivors/carers availing of this. One carer added some minor clarifications, and these were recorded in field notes.

ES received coaching in best practice when carrying out interviews with people with cognitive and communication problems. Strategies to support communication were tailored to the participant, and included sharing the interview questions in advance, using clear and simple wording and allowing the person adequate time and space to respond.

The data collection period coincided with a time of varying levels of public-health restrictions due to Covid-19. Interviews were therefore conducted remotely by telephone or MS Teams, with the exception of one interview with a senior clinical leader which took place in-person and masked. The survey was conducted using the online Qualtrics platform.

*Analysis*. Interviews were transcribed and responses to the open-ended online survey questions were collated, and imported into NVivo (Version 12 QSR international) software. Framework analysis was used to analyse the data, following the approach outlined by Gale et al. [18], and informed by Braun and Clarke's thematic analysis approach [28]. A broadly interpretivist, reflexive approach was taken to the analysis, with an acknowledgement of the role of the researchers in shaping the analysis and the findings [29].

The first stage of analysis was familiarisation, involving a close reading of data and independent open coding by two coders (ES and KF) of transcripts from the first five interviews completed. Tentative top-level categories were developed and continually revised in an iterative and collaborative process. The categories were informed by conceptual frameworks related to patient navigation [30, 31] and stroke care domains in the Action Plan for Stroke in Europe [22].

The resultant working analytic framework was independently applied by both coders to three further transcripts. Differences were discussed, resulting in further revision of the analytic framework. Once agreed, this framework was applied deductively to the remaining transcripts and survey responses, divided between the two coders.

Following this, the data was charted into a framework matrix by ES. This allowed the data to be summarised for each interview and survey response, by the categories in the analytic framework. The final stage of framework analysis was interpretation, which focussed on identifying specific priorities for improving stroke care, and specific improvements suggested by stakeholders for each of these priorities. The results of this analysis were shared with 3 survivors and 1 carer for comment, with no changes made.

The next stage involved mapping potential care improvements to these priority areas, drawing on both the qualitative data collected within the study, and key policy documents [8, 22, 32]. This resulted in a long-list of potential improvements, which was reviewed by members of the steering group, and further revised based on their review.

**Phase 2: Priority setting survey.**   In Phase 2, participants completed a survey where they were asked to choose five priority stroke improvements from the long-list generated in Phase 1. This is the simplest ranking format recommended by JLA, and was selected as the most inclusive for all participants. The list of improvements was grouped by service area for readability, and the order of presentation of each service area group was varied at random across participants. The survey was piloted with 5 participants (1 survivor, 1 carer, 2 healthcare professionals and 1 researcher), resulting in minor revisions to instruction and question wordings. Participants also provided high-level demographic data, and could provide open-ended feedback. The survey questions are available at the OSF link above.

Participants had the option of taking part online, by completing and returning a paper survey via post, or via a facilitated interview format over the phone or face-to-face. The online survey was programmed and administered using the Qualtrics platform. In the facilitated interview format, the participant responded to survey questions that were identical to the online version, which were provided in advance. They also had the opportunity to discuss the long-list with the interviewer (ES), who provided clarifications and reminders where necessary, while taking care not to influence the participant's choices. Only the responses to the survey questions were recorded, via Qualtrics. Interviews lasted between 30 and 90 minutes.

*Analysis*. For each improvement we calculated the percentage of participants in each stakeholder group that had selected it as one of their top five. To ensure a sufficient sample size, survivors and caregivers were combined into a single group for this purpose. A top priority improvement was defined as an improvement that was selected by at least 20% of professionals, or at least 20% of the combined survivor/caregiver group.

**Stakeholder meeting.**   In preparation for the stakeholder meeting, a candidate list of improvements was drawn up. The criteria for this list were broader than those outlined above

for a top priority, and included any improvement that was selected by at least 20% of any of the three stakeholder groups, rather than combining survivors and caregivers. This ensured that any improvements prioritised by carers could be considered.

Some improvements were excluded from consideration, as they were not suitable for evaluation using modelling. This included, for example, improvements where it was difficult to quantify the benefit for stroke survivors or their carers, or where there were no relevant, evidence-based interventions. Evidence of effectiveness was identified for each candidate improvement by searching the Cochrane database: https://www.cochranelibrary.com/cdsr/about-cdsr. At the outset of the research, we had pre-specified criteria for the final set of improvements (see protocol for more detail [15]). At least 4/5 improvements would have RCT evidence of effectiveness, and 3/5 would have systematic review evidence. There would be one improvement from each of the three phases of care–prevention, acute and community.

Participants were provided with advance information one week prior to the meeting. Loosely following the Nominal Group Technique approach [33], the meeting involved a series of structured discussions, followed by a vote. This approach allowed for both efficient decision-making and consideration of everyone's opinion. The meeting was not recorded, but anonymised minutes were taken, circulated for review and agreed following the meeting. Survivor and family carer participants were reimbursed for their time. The meeting took place online using the Zoom platform.

## Results

### Phase 1 –Initial priority gathering—Participants

In-depth interviews were conducted with 18 survivors (8 women, 10 men, age range 39–81) and 13 family/main carers (12 women, 1 men, age range 34–76). There was a total of 29 survivor/carer interviews—two were carried out with survivor/carer dyads, and 27 were individual interviews. Interviews were conducted with 8 senior leaders (6 women, 2 men). Of the 37 total interviews, 23 were phone interviews, 13 Microsoft Teams calls, and one face-to-face. Interviews lasted between 30 and 130 minutes, with an average length of 69 minutes. One interview was split over two occasions to accommodate interviewee fatigue. There were 88 responses to the online survey– 80 professionals working in stroke care, 5 family/main carers and 3 survivors.

One survivor and one carer who initially provided contact details did not respond to follow up phone calls, and one survivor and one carer declined to participate in an interview following review of the participation information leaflet. None of the interview participants who signed and returned a consent form withdrew from the study. In addition to the 8 senior leaders who completed interviews, a further 10 were invited, with no response received from 8, and 2 declining to take part. For the online survey, 47 participants consented to participate but did not respond to any of the survey questions. Of the 88 eligible responses who answered at least one question, 33 answered 9/9, 39 did not answer 1 or 2 questions, and 16 did not answer more than 2 questions.

S2–S5 Tables display the participant characteristics in detail. All geographical areas of Ireland were represented within each stakeholder group. Among the professionals responding to the survey, 49/80 (61%) classified themselves as an allied health professional, and 21% as another health professional (e.g., doctor, nurse). The remainder included researchers, managers and voluntary sector staff. The most common service area was acute services (45/80, 56%), followed by rehabilitation services (24/80, 30%).

**Phase 1 –Initial priority gathering–Results.** The final analytic framework consisted of 13 higher level categories. Six related to care domains–pre-hospital care, acute care, rehabilitation,

primary prevention, secondary prevention and life after stroke. Four categories related to navigation–the healthcare environment, navigation behaviours, individual characteristics, and provider factors. There were three cross-cutting categories–national policy, quality of care, and social support. Within these higher-level categories, there were 37 sub-categories. These were mapped to themes related to 15 priority care areas, grouped within seven broader areas. These seven broad priority care areas are described briefly below, and in Table 1. Detailed descriptions of the analytic framework, including the 13 higher-level categories, 27 sub-categories and corresponding codes, and sample quotes are available at the OSF link above.

**Acute care–timely access to high quality care and a good inpatient experience.** Survivors and carers generally reported excellent experiences with acute treatment. Professionals more frequently mentioned a need to ensure faster access to acute treatment for more people, through things like improved pre-hospital processes and re-configuration of services. The FAST criteria (Face, Arm, Speech, Time) were well-known among the participants as a way of recognising the symptoms of stroke. However, those who had a less typical stroke presentation felt that this delayed their access to treatment, with one survivor noting how "*FAST didn't work for me really.*" [S_14].

The acute setting was often described as an unpleasant environment to be in. Several participants mentioned problems with noise and difficulties sleeping, and a general lack of attention to individual needs. Examples of this included not being able to get vegetarian meals, being told not to use a mobile phone after a certain time, as well as more serious issues such as call bells not working, or issues related to pain management.

**Communication and navigating the system.** The importance of good communication and support for navigating the care system was mentioned frequently by all stakeholders, but particularly by survivors and carers. One carer said that "*the biggest gap I think was the communication*" [C_12], and this view was shared by most participants. Key information gaps included 1) information on the cause of the stroke; 2) information on what to expect after the stroke, including things like low mood or fatigue (*"I didn't realise that, like the fatigue, like I'm floored with fatigue"*, [S_04]) and 3) where to access help and support.

Many survivors and carers discussed the difficulties of accessing staff to ask questions in busy, high pressure acute settings. One carer [C_12] described how she felt she had to "*stalk*" the consultant. A survivor with aphasia (S_14) described significant communication difficulties in the acute setting, a situation exacerbated by not having any family support during Covid-19 restrictions–"*I couldn't speak at all . . . I know* [the hospital staff] *were under awful pressure and everything. But I felt that they could have tried a little bit harder to perhaps understand.*"

Navigating the care system was experienced as extremely challenging. This was particularly the case at the time of discharge from an acute setting or rehabilitation—"*I left, and moving onto the next step I was left. I had no . . . nothing*" [S_07]. Participants often felt that they had to "*fight*", "*push*" or "*badger*" to secure needed services, which was time consuming and a source of stress. There was no standard post-discharge care pathway, with variation in service provision around the country. Participants often described how they "*stumbled across*" [C_10] a service, or identified the right service through "*sort of trial and error and foraging around*" [C_05]. Eligibility criteria were often seen as unclear and arbitrary–in particular, there were often age criteria for services and supports, with older adults unable to access more specialized services, and younger adults not entitled to support for equipment or housing adaptations.

*"This is what really shocked me, [the neuro-rehabilitation service] said, 'Sorry we don't treat anybody over sixty-five.' . . . Sure, that's just crazy . . . It's quite an insult actually to be honest with you . . . You know sixty-five then you're ready for the heap." [C_05]*

**Table 1. Phase 1 priority areas of care based on qualitative responses from survivors (n = 20), carers (n = 18) and professionals (n = 80).**

| Phase 1 Priority Areas of Care | | Survivor Quotes | Carer Quotes | Professional Quotes |
|---|---|---|---|---|
| **Acute Care** | Timely access to the highest quality acute care possible | "when it happened, I got super care because I got the thrombolysis . . . I wouldn't be here otherwise" [S_04] | "there was a lot of delays in getting my mum to hospital because it just developed really, really gradually . . .very kind of like slow, insidious didn't fit the picture, and because it didn't fit the picture it didn't sort of trigger the same alarm bells." [C_04]" | "the speed of access, the quality of access, and the equality of access for everybody" [SL_01] |
| | Person-centred inpatient experience (e.g. sleep, food) | "it took six weeks for them . . . that I'd get the right [vegetarian] dinner." [S_16] | "his wheelchair could barely fit between the bed and the window . . . television left on sky high". [C_11] | N/A |
| **Communication and navigating the system** | Good quality communication and information | "I did find lacking was the explanation of what happened . . . Because it's part of the rehab . . . is for me anyway is understanding what has happened." [S_03] | "Communication, communication or lack of communication was to me, was desperate". [C_05] | "in a hospital, people think, they're too busy and they are too busy for that kind of delayed follow up, and the chatting to the patients . . . so it's the system that lets people down." [SL_01] |
| | Navigating the system—identifying and accessing needed services | "it's a full-time battle to make sure that you get, I'm only looking for what I'm legally entitled to." [S_06] | "[a physiotherapist] led us to chase things that we didn't know about. And they're basic. . . when you know what you're looking for, you can go and find it . . . like fittings for her shoes, like the use of a commode, like the use of a side on the bed, any of those things." [husband of S_12] | "a lot of mood problems are created by disjointed systems. Where people just don't know . . . how to navigate in terms of the complexity to . . . It's those practical things that exacerbate the stress." [SL_06] |
| Rehabilitation | Maximising rehabilitation potential for everyone | "there didn't seem to be great facilities in the community . . . from being in hospital getting physiotherapy 5 days a week, sometimes twice the one day, I came to the situation where I was going to have it every 5 or 6 weeks." [S_11] | "Yeah, there's a sort of a sense that if you're a certain age, you're, you know, you're not going to rehabilitate but, you know? Everyone can rehabilitate. It's just about changing the goal posts to where they need to be." [C_04] | "[services need to be] delivering an adequate dosage of therapy towards meaningful goals." [SL_07] |
| | Support for neglected outcomes–cognitive, mental health, communication, fatigue | "this has made me cry several times. I'll do something like [a training course] and I'd be really involved in it. And then it doesn't stay in my brain like it is, it's gone." [S_03] "I was able to talk to somebody . . . that made a big help, you know, it made a big difference to my life, like, it really did." [S_01] | "the main thing I would have, I would say is lacking is probably looking at the brain, you know, the effect on the brain as well as on the body" [C_09] "I did counselling [a long time after the stroke] . . . And I found, I should have done this years ago for myself, for the way things had kind of changed in my life with him, because I found it really good, really, really good". [C_09] | "Ensure access to SLT not only for dysphagia support (as is commonly prioritised) but for communication support." [Survey] "that's something definitely we at the moment in Ireland, we don't have enough counselling services. I don't think for anything really." [SL_03] |
| Prevention | Risk factor awareness, screening and health checks | "my, my one thing [that the government could do] was to have GPs check everyone's BP [blood pressure]" [S_04] | "there should be much more awareness that stroke is not this big dramatic one-off event. You can have many mini-strokes or little strokes or things like that beforehand . . . if people were more aware of it, people, you know, could move faster." [C_02] | "[we need] a multi-pronged approached . . . educating GPs on screening for stroke risk factors, access the HSCP therapist, smoking cessation etc. Policies: national policies prioritising mental health, stress reduction, healthy eating and physical activity" [Survey] |
| | Evidence-based support for healthy lifestyle | "I was not given a great deal of guidance on keeping healthy after the stroke, although the Physiotherapist did give some guidelines on physical activity." [Survey] | "my question the whole time was for the doctors. sometimes you'd feel they weren't really listening because again I wanted to know why did he have another clot. In other words, how could we prevent a further one."[C_06] | "[Medications is] where all the thought process has gone. It hasn't gotten beyond a tablet." [Survey] |

*(Continued)*

**Table 1.** (Continued)

| Phase 1 Priority Areas of Care | | Survivor Quotes | Carer Quotes | Professional Quotes |
|---|---|---|---|---|
| Life After Stroke | Return to driving, work, purposeful activities, social connection | "for me to be scared to go out to the car and drive down the town in [local town], that has a huge burden on me." [S_03] | "[Work] was good for his mental kind of place to be in, you know. Kept his brain going and kept an interest in things" [C_09]<br>"He needs more than me to speak to, he needs other men and you know just to talk over things." [C_01] | "[We need] broad thinking on what constitutes therapy—include family, work and social settings and contacts, employers (where necessary) and community groups working alongside traditional health professionals". [Survey] |
| | High quality, accessible long-term care | "I suppose people might have a fear of nursing homes a bit as well. And kind of demystifying it a bit you know and sort of saying, actually this can be a pretty positive place to be in." [S_19] | "The discharge co-ordinator then like just wanted you to take whatever option was available" [C_12] | "this group is so often overlooked as they are 'safe' but bored and under-stimulated, leading to other health and social issues." [Survey] |
| | Support to stay independent at home | "some of [the home carers] were lovely like, were really nice and all but you just, you'd kind of bond with them, and then they'd never come back". [S_04]<br>"I'm doing all my own care of the house, but again I just run out of steam." [S_14] | "a lot of the so-called community care, kind of involved getting the equipment right, and off you go." [C_11] | "[the] person with stroke is showered but has little access to other activities in life." [Survey] |
| | Peer support and peer-led advocacy | "I found online the only place to get good info—I wanted to be inspired by other stroke survivors and to know my life wasn't over." [Survey] | "I couldn't praise that [stroke support group] service enough." [C_06] | "Moving education and support away from clinicians to people who have lived experience of this and can support without being directly responsible for their care and rehab" [Survey] |
| Support for carers | Involvement of family members in care | "what . . . could've been better was family contact. Now I know it was at the height of COVID. I didn't really have much contact with the family. Partly I suppose because I couldn't work my mobile phone or anything. I wasn't able to." [S_14] | "nobody told you what you could do to help [in the acute setting], do you know it was kind of, and I suppose we never knew what was happening, we didn't realise he was as unwell as what he was I think." [C_12] | "So sometimes families they're just not, they're not included as partners in care. You know what people don't realise is that what happens to a person happens to a family. And that has not hit the culture of care." [SL_05] |
| | Support structures for carers | "The biggest problem I felt was there wasn't much given to [my wife] or [my son]. Do you know in the sense of rehab, and the programmes they have out there, there was nothing really given to the caregivers or to the families. They don't get support, like you know." [S_02] | "I mean you're on your own, you really are, you're on your own." [C_01]<br>"I didn't have time for support really . . . You know, I suppose it's um, it's like when you get someone to clean your house. You have to clean it first before they come." [C_11] | "carers health, both mental and physical, can be impacted due to being left alone to get on with it . . . They need reminding to firstly look after themselves so that they can look after their loved one." [Survey] |
| Awareness, research and policy | Awareness of stroke in general public and healthcare professionals | "People you might see you from the outside and think you're grand. And but everything you know that like the problems that you're having are hidden then, yeah." [S_15] | [Speaking about her brother with aphasia]—"[we needed] an advocate to say you know what, he's cognitively intact, he just can't understand you. . . some of [the healthcare professionals] treated [him] like he had a cognitive impairment." [C_10] | "People aren't aware enough about the spectrum of impacts and outcomes from stroke, leaving a major vacuum in opportunities for someone who has experienced stroke and their family for info and peers." [Survey] |
| | Supportive policy, workforce, research and data environment | "[we need to] improve monitoring, greater understanding, research." [S_14] | "do you know actually what was huge up [in the nursing home], I mean the physio was great and all the rest of it, the attention, sure they're the experts in the field." [C_12] | "So, there's a problem between policies, frameworks and implementation and frankly nothing's been implemented." [SL_07] |

*"If I was older . . . we would have had the house kind of kitted out."* [S_04]

Survivors and carers valued when healthcare professionals took the time to pro-actively check in with them, and to provide clear and honest information. One carer mentioned that she was given a "*roadmap of what would happen*" [C_10], but felt that she likely got that because she was an experienced health professional herself, and knew what to ask for. Some participants experienced very valuable support for navigation from professionals such as local stroke support group co-ordinators, therapists or general practitioners, but access to this type of support was variable across participants.

**Rehabilitation—Maximising rehabilitation potential for everyone and support for neglected outcomes.**   Survivors and family carers were keen to maximise their recovery, and many felt that they would have benefited from more hours of therapy, over a longer period of time. Some expressed disappointment in the level and intensity of therapy in inpatient rehab settings–*"I thought I was going to be in this place where I'm going to be, you know, pushed from, get going, you know, but I wasn't."* [S_07]. Others were happy with the level of therapy in inpatient settings, but disappointed in the level provided post-discharge. Professionals also believed that the hours of therapy given are often not sufficient to maximise recovery. Several participants used privately provided therapy services to supplement therapy provided through the public system.

The need for access to specialist rehabilitation, closer to home, was mentioned frequently. Survivors often had to choose between staying close to family, which they saw as essential for their recovery, and travelling for specialist rehabilitation. There were also waiting lists for specialist care, and age criteria for access. Many professionals working in stroke care mentioned the need for standardised and holistic assessment of need, to ensure that access was not based on the subjective assumptions of clinicians, or arbitrary criteria such as age.

Survivors and family members particularly valued rehabilitation that was tailored to their specific goals, preferences and capabilities, and with flexible timing, rather than a "one size fits all" approach. As one carer said–*"They don't talk to the patients enough or look at the individual"* [C_02].

The lack of attention paid to outcomes other than physical function in rehabilitation was mentioned frequently across stakeholder groups. This included cognition, mental health, communication and fatigue. Almost all survivors and carers spoke about having "*dark days*" [S_02] of low mood and depression after stroke, and also about the huge psychological impact of the stroke on the entire family–*"You know it alters everyone's life and your life is forever altered going forward."* [C_06]. Several survivors and carers had good experiences with counselling or psychotherapy to help with these issues, but there were barriers to accessing this type of support. These included time (especially for carers), stigma, and the need to find a counsellor that was a good "*fit*" for them or that "*understood.*"

**Life after stroke–securing safety, independence and purpose.**   Life after stroke priorities included 1) driving, work, purposeful activities, and social connection, 2) support to stay independent at home, 3) high quality, accessible long-term care, and 4) peer support and peer-led advocacy (see Table 1).

Many survivors and carers emphasised the importance of support for driving and returning to work, where this was feasible. The process for achieving these goals was often difficult to navigate, with challenges accessing accurate information and support. One healthcare worker involved his union to facilitate a gradual return to work–*"the union wrote to them and basically put it to them that they had a duty to me to aid my recovery . . . I [went] back to 12 hours a week . . . was increased to 24 hours and then it was increased to 39"* [S_11].

Opportunities to take part in meaningful, social and enjoyable activities were seen as very important. This included activities like volunteering, hobbies, gardening, and arts activities

such as music, drawing and dance. As one survivor described, "*I began to realise, you know, that I wanted to live a happy life after stroke.*" [S_02].

Support to stay at home was seen as crucial, particularly for survivors with more severe disability. Home care hours were seen as essential for this group, but quality and availability were mixed, and did not always fit with the specific needs of the survivor. For example, home care was only available for self-care needs, but many survivors needed help with housework, or doing physiotherapy exercises. Respite and day services were seen as very helpful for both carers and survivors, but difficult to access and very time-limited–"*a drop in the ocean*" as one carer put it [C_09]. Support for equipment and housing modifications was also seen as essential, but difficult to secure, with complex and unclear eligibility criteria. One carer described how the home carer was unable to shower her husband as the bathroom was upstairs and inaccessible. Although they could get a grant for a downstairs bathroom, it wouldn't cover the full cost–"*[the home carer] had no facilities, she couldn't wash him; she could do nothing . . . [the] house isn't suitable . . . there would be a certain amount from the . . . housing grant area . . . but then only a percentage.*" [C_08].

Many survivors and family carers were reluctant to consider long-term care (LTC). It was perceived as isolating and lonely, with minimal stimulation. The process of obtaining a place in LTC was also experienced as complex and difficult to navigate. On the other hand, of the survivors and family carers who had experience of LTC settings, the experiences were often quite positive. One major issue identified was that stroke survivors living in LTC do not have the same entitlements to therapy and equipment that those living at home do. One stroke survivor LTC resident described this—"*I can't even get a bog standard [wheelchair] . . . they just won't give it to me. Because I'm in a nursing home.*" [S_19].

Stroke support groups were a valuable source of social engagement, emotional support, activity and stimulation, and a break for carers. One family member described a stroke support group as "*one of the best things I've seen out of anything [my husband] has been offered*" [C_09]. Opportunities for peer leadership and advocacy were seen as important–survivors were keen to share their knowledge/experience with other survivors, which built their confidence and sense of purpose. One stroke survivor emphasised the value in sharing lived experience —"*[other survivors] have the experience. Whereas you know professionals haven't been through it*" [S_10].

### Primary and secondary prevention

All stakeholders identified awareness and understanding of stroke risk factors as a key priority for prevention. Stroke survivors and carers described missed opportunities to prevent a stroke, including failures to detect high blood pressure, intermittent atrial fibrillation (AF), or recognise the symptoms of or adequately manage a transient ischemic attack (TIA). Less well known risk factors such as sleep apnoea and stress were mentioned frequently.

Professionals working in stroke care emphasised the need for structured, evidence-based secondary prevention programmes, involving education, advice and support for behaviour change. In general, survivors found support for lifestyle change fairly limited, with advice tending to be general, with "*nothing concrete*" as one survivor [S_03] put it. Gym access, adapted exercise equipment and exercise classes were identified as helpful for staying active. An adapted bike provided by local community services was highlighted by three participants (C_09, S_17 and C_02) as very helpful for staying active, however this bike was provided only for a limited time due to a shortage. One survivor paid for a similar bike out of pocket following this, and saw it as essential to staying active—"*I was never going to improve if I didn't use that you know.*" [S_17].

### Support for carers and carer involvement

Carers described supporting survivors in multiple ways—acting as advocates, supporting navigation, meeting basic care needs, supporting recovery and prevention, and social and emotional support. Many carers felt isolated and unsupported as they cared for their loved one, while also adjusting to the changes to their lives as a result of the stroke. Despite this, some carers found it difficult to access support, with many carers saying they did not feel "ready" for counselling or felt they could not prioritise their own mental health–"*I didn't have time for support really*" [C_11].

Family carers often did not feel involved enough in the survivor's care–they were not kept sufficiently informed, did not have opportunities to be involved in therapy, and did not feel that they had sufficient training. Carer involvement was particularly poor during Covid-19 restrictions.

### Awareness, research and policy

Survivors and family carers often encountered health-care staff with limited understanding of post-stroke cognitive and communication difficulties, and how to best support individuals with these problems. Friends, extended family, and the general public also often had limited understanding of stroke, particularly hidden disability such as problems with fatigue, cognition or communication.

Professionals working in stroke care highlighted the need for implementation of existing policies and strategies, and improved routine data collection, audit, and research. The need to recruit and retain more specialist stroke staff was a frequent concern among professionals, and seen as a necessary pre-condition to improving services.

Stroke survivors and family carers identified a need to be involved in service development —"*it's about bringing resources, and it's about bringing knowledge, and it's about bringing I suppose the capabilities to the table*" [S_02]

### An over-arching theme of under-resourcing

A common thread or over-arching theme running across all seven priority areas was a general scarcity of staff and resources to match the needs of the stroke population. This was reflected in waiting lists for needed services and pressure on staff leading to poor communication and information provision. It was notable that communication was often experienced as better in rehabilitation settings, which tended to be less busy and pressured relative to acute settings. Navigation challenges also appeared to be linked with this underlying scarcity of resources. For example, one key barrier to effective navigation was the use of unclear or arbitrary eligibility criteria for services, such as age limits—one senior leader described this as a "*crude*" way of "*rationing*" services that were limited in supply [SL_08]. The variation in service provision around the country can also be linked with scarcity, as gaps in services have been filled by small-scale, local-specific initiatives developed by local champions and voluntary organisations, resulting in what one senior leader (SL_04) described as a "*patchwork*" of services.

### Long-list of improvements

The seven priority areas of care were mapped to a long-list of 45 potential improvements. There were 6 improvements identified for acute care, 7 for communication and navigation, 8 for rehabilitation, 10 for life after stroke, 7 for prevention, 2 for carer support, and 5 for awareness, research and policy. This complete long-list is available at the OSF link above.

## Phase 2 –Priority setting survey—Participants

The survey was completed by 95 participants: 42 professionals (all online), 34 survivors (28 online) and 19 family carers (17 online). Two family carers and three survivors completed the survey by post. Three survivors completed the survey via facilitated interview, two by phone and one in-person. S5 Table displays the characteristics of the Phase 2 survey participants, with 2/17 family carers and 8/33 survivors aged 65+, and 5 survivors aged 70+. Across groups, the highest number of participants were Dublin-based, but all geographical areas of Ireland were represented. Among professionals, there was a mix of profession and service area.

## Phase 2 –Priority setting survey—Results

There were five priority improvements that were selected by at least 20% of the participants as a whole (see Table 2). These were 1) improved access to community neuro-rehabilitation, 2) a

**Table 2. Phase 2 top priority improvements selected by professionals, survivors/caregivers and overall.**

| Priority Improvement | Additional detail | Professionals (n = 42) | | Survivors and Caregivers (n = 53) | | Total (n = 95) | |
|---|---|---|---|---|---|---|---|
| | | N | % | N | % | N | % |
| Access to specialist community neuro-rehabilitation. | Adequately staffed 7-day multi-disciplinary team Standard rehabilitation needs assessment, no age criteria Sufficient number of hours and intensity to ensure benefit | 17 | 40 | 13 | 25 | 30 | 32 |
| Comprehensive community stroke support service providing long-term access to meaningful, social and beneficial activities and programmes | Access to a range of meaningful, enjoyable and social activities Standard needs assessment for each survivor Group-based and individual activities and classes–e.g., exercise classes, self-management, fatigue management Referral to other community-based services | 13 | 31 | 16 | 30 | 29 | 31 |
| Investment in recruitment and retention of stroke specialist staff. | Government commitment to meet staffing targets. Regular audit of staffing levels and targets. Improve working conditions for staff retention. | 11 | 26 | 12 | 23 | 23 | 24 |
| Comprehensive community stroke support service, to provide information and help with accessing services | Signposting and help with accessing services Nurse helpline for advice Information and group education sessions, online or in-person Available for as long as needed | 9 | 21 | 13 | 25 | 22 | 23 |
| Access for as many stroke patients as possible to specialist stroke acute care | Improve access to relevant acute treatments including thrombolysis and thrombectomy Fully staffed multi-disciplinary specialist stroke care, and early rehabilitation. Ensuring stroke units meet relevant standards (e.g. European Stroke Organisation) | 9 | 21 | 10 | 19 | 19 | 20 |
| Explore ways of improving the speed of access to stroke treatment for people with less typical presentations (i.e. those who do not fit the FAST profile). | Currently there is no clear evidence-based way of improving speed of access for non-FAST strokes. There is a need to explore ways to do this that would also balance the risk of overwhelming stroke services with people who are not having a stroke. | 5 | 12 | 13 | 25 | 18 | 19 |
| Improve access to specialist inpatient rehabilitation. | Adequately staffed 7-day multi-disciplinary team Standard rehabilitation needs assessment, no age criteria Sufficient number of hours and intensity to ensure benefit | 10 | 24 | 6 | 11 | 16 | 17 |
| Improve access to early supported discharge services | Adequately staffed 7-day multi-disciplinary team Standard rehabilitation needs assessment, no age criteria | 10 | 24 | 4 | 8 | 14 | 15 |
| Added in stakeholder meeting: Improve access to support for mental health difficulties after stroke | Mood screening to identify survivors in need of support. Access to anti-depressant medication where required. Access to specialist clinical psychological or psychiatric support where needed | 5 | 12 | 8 | 15 | 16 | 17 |

comprehensive community stroke support service, 3) investment in staff recruitment and retention, 4) a service to provide information and support with navigation, and 5) improved access to stroke specialist acute care.

Within the stroke survivor/carer group, 25% prioritised exploring ways to improve access for strokes with atypical presentation. Among professionals, 24% prioritised specialist inpatient rehabilitation, and 24% prioritised early supported discharge. No improvement within the prevention area of care was selected by at least 20% of any group. The full results of the survey are available on OSF.

## Stakeholder meeting

20 participants were invited to take part in the stakeholder meeting, with 10 agreeing to participate and attending on the day (4 survivors, 2 family carers, and 4 professionals working in stroke care). The professionals included a clinical nurse specialist, a speech and language therapist, a clinical neuropsychologist and an occupational therapist. A further two study steering group members participated in the meeting and the voting–the study PPI representative, and an academic researcher (not the lead researcher).

Prior to the meeting, a candidate list of priority improvements was compiled, corresponding broadly to the top priorities listed in Table 2. Additional priorities were included that were selected by either at least 20% of survivors or at least 20% of carers. This included, for example, support for mental health difficulties (>20% of survivors) and support for carer involvement (>20% of carers). A number of these additional priorities related to information and navigating the system, and these were combined into a single improvement of services for providing support with information and system navigation.

"Exploring ways of improving access for atypical presentations" was excluded from consideration for evaluation with modelling, due to insufficient evidence for effectiveness of potential interventions, and an absence of any relevant evidence-based guidelines. "Investment in recruitment and retention of stroke specialist staff" was excluded from consideration due to difficulties quantifying a direct benefit for stroke survivors or their families. These exclusions were discussed and agreed in the meeting.

In order to maintain the focus on stakeholder priorities, it was decided to disregard the pre-specified criteria that required a prevention improvement would be included, as none met the criteria for the candidate list. All of the candidate improvements had RCT evidence of effectiveness, and almost all had systematic review evidence. This included, for example, Cochrane reviews providing evidence for the effectiveness of community rehabilitation [34, 35] and information provision [36]. The list of candidate improvements, and the advance information provided to meeting participants, including information on effectiveness, is available at the OSF link above.

The general meeting discussion echoed the Phase 1 findings. Meeting participants emphasised that services need to be well-organised and co-ordinated, stroke-specific where possible, and national and consistent, with the same services provided everywhere. There was broad consensus in the discussion around the final top 5 improvements, and this was also reflected in the voting. These were 1) Specialist community neuro-rehabilitation (12/12, 100%), 2) Services to provide information and support access to services (11/12, 92%), 3) Community support service with long-term access to meaningful, social and beneficial activities and programmes (9/12, 75%), 4) Specialist acute care (9/12, 75%) and 5) Support for mental health difficulties (9/12, 75%).

These aligned broadly with the top priority improvements identified in Phase 2, with the exception of support for mental health difficulties, which had been slightly below the threshold for a top priority, but which meeting participants felt was important to include (see Table 2).

Family carer involvement was not selected as a final priority, but participants emphasised that this should be a key consideration across services. S1 Fig shows how priority improvements were identified at each stage of the research.

## Discussion

The top priorities that stakeholders identified for improving stroke care in Ireland were 1) improved access to specialised neuro-rehabilitation (particularly in the community), 2) better ongoing support for life after stroke, and 3) improved information provision and support for navigation of services. Staffing, specialist acute care, speed of access for atypical stroke presentations, and support for mental health, were also identified as priorities. There was a common over-arching theme of services needing to be more standard and consistent nationally, and insufficient resources (e.g., clinician and home care time, funding for equipment) to meet the needs of stroke survivors and their families.

There were some key differences across stakeholder groups. Professionals demonstrated awareness of issues with navigating services, but did not mention communication and information as frequently. Professionals did not mention the quality of the inpatient experience (e.g., noise, sleep, facilities).

Professionals also did not mention issues related to speed of access for atypical stroke presentations, or prioritise this in Phase 2, perhaps due to an understanding of the difficulties involved in addressing this. In Phase 2, a higher proportion of the professional group prioritised early supported discharge (ESD). Carers and survivors may have had less awareness of ESD as a potential option, as it has only become widely available recently [6]. Inpatient rehabilitation also had a higher priority among professionals, potentially due to survivors and carers preference for rehabilitation closer to home, which was apparent from the interviews.

The need to improve communication, information provision and the ease of navigating services post-stroke, are consistently identified as issues in studies of stakeholder priorities for stroke care [11–13]. A recent study in New Zealand found that "knowing what is available" is a key barrier to accessing services, a concern echoed by many of our participants [13]. Inadequate post-discharge follow-up care was also identified as an issue in two very recent studies [12, 13], consistent with the priority placed on community and longer-term supports in this study. The authors of a meta-ethnography in this area [11] noted the consistency in concerns among stroke survivors and carers across diverse national contexts and health systems, and our study supports this.

Previous research, including Irish-specific research, has found that psychological support is a key priority for stroke survivors [12, 14]. Although support for mental health difficulties did not meet the 20% threshold for a top priority in the survivor/carer group combined, it was selected by 21% of survivors, and in the stakeholder meeting it was agreed that it should be included as a top priority improvement.

There were some issues raised in this study that have not been prominent in previous research. This included problems with the inpatient experience which may be a particular issue for Ireland, and may have been exacerbated during the Covid-19 period. However, similar issues were experienced by survivors in this study who had their stroke pre-Covid. The need to improve speed of access to treatment for non-FAST strokes has not been highlighted previously, but is unlikely to be an Irish-specific issue. This may have been identified in our study due to the open and in-depth nature of the enquiry.

### Implications for policy, practice and future research

There is an urgent need to implement specialised community neuro-rehabilitation for all areas of the country, accessible to all age groups, at a sufficient level of intensity, and based on a

standardised assessment of need. This is a key element of the managed clinical rehabilitation networks that are envisaged for Ireland [7]. This finding supports the increasing international emphasis on the need for greater hours and intensity of rehabilitation post-stroke, evidenced by recent UK guidelines [37]. The guidelines also emphasise that rehabilitation must be tailored to individual needs, consistent with our findings. Recent research suggests that variability of individual needs (along with resource constraints) can complicate guidelines-based delivery of rehabilitation, indicating that a "one-size-fits-all" approach is not feasible [38].

Findings from this study also support an emerging recognition internationally that time-limited neuro-rehabilitation is not sufficient, and that stroke survivors need longer-term support, for this key "Life After Stroke" phase [22]. There is no current consensus on an optimal model of care for this phase of care, although there is work ongoing [39]. Important elements likely include evidence-based therapies such as self-management [40], and that address key areas of unmet need such as fatigue and emotional support [14]. These types of services are currently offered in Ireland by voluntary organisations, but in a fragmented, piecemeal fashion, and need to be offered and publicised in a consistent way around the country.

Issues related to service navigation and communication are mentioned in key stroke national and international policy documents [22, 32], but are arguably not addressed at a level of detail that reflects their importance to stakeholders. The findings from this study indicate that a comprehensive plan is needed for information provision for stroke, optimally co-designed with stroke survivors and families. This would involve active information provision [36] at multiple time-points, in multiple formats, tailored to the individual, and with family members and carers included.

Delivery of stroke services is complex due to the range and individual nature of post-stroke deficits, and as such requires both specialisation of tasks and co-ordination across tasks. Healthcare systems often prioritise specialisation at the expense of co-ordination [31], and this may explain the frequent challenges encountered internationally with care continuity and patient navigation. The solution to this likely lies in both services to support patient navigation (e.g. individual key workers, group-based supports) and clearer, more standard and transparent care pathways that are easier to navigate [31].

There has been considerable focus on reducing pre-hospital delay in stroke policy, with the emphasis on detection of FAST-positive cases (a mnemonic for the public for **f**acial drooping, **a**rm weakness, **s**peech difficulties; **t**ime to call emergency services). However, the need to detect non-FAST cases has been relatively neglected in research. Tools such as BE-FAST broaden the criteria to include balance and visual symptoms [41], but do not include non-traditional symptoms such as mental status change which can be frequent particularly in women [42]. Many people with milder strokes, including those with "non-disabling" symptoms, can experience barriers and delays in treatment, despite the fact that these strokes can be associated with longer-term neurological deficits [43]. Further research is needed in this area to help the general public and healthcare professionals identify strokes associated with non-traditional stroke symptoms quickly and earlier, without the risk of casting the net too wide and overwhelming services with stroke mimics. Machine-learning is one approach that may show promise in this area [44].

The study findings indicate a general pattern of under-resourcing along the stroke care pathway, with stakeholders recognising particular deficits in staffing. There is a need for greater understanding of the level of demand for stroke services in Ireland, and the resource required to meet that demand. Policy-makers in Ireland have emphasised the need for population-health-based planning to inform service decisions [45]. A key next step of the current project will be to use population-based modelling to estimate current and future need for stroke services in Ireland [5, 15], and evaluate alternative policy and service options in terms of cost and cost-effectiveness.

## Strengths and limitations

A key strength of this study has been the combination of rich qualitative data with more specific quantitative data. The scope of this study was more open and less pre-defined than previous qualitative studies, allowing participants to highlight issues that may be less well known to researchers and clinicians. We used an inclusive approach to recruitment and data collection at all stages to facilitate inclusion of participants with cognitive and communication difficulties. Limitations include the small sample size, particularly for caregivers in the Phase 2 survey. Caregivers were a particularly difficult group to recruit, perhaps because they are uniquely busy, and may not be as engaged in relevant organisations. This likely meant that carer priorities were not adequately represented, and that the needs of people with more severe stroke were under-represented. Another key limitation was the absence of any prevention improvements in the final priority list. The focus on perspectives and experiences of stakeholders may have led to prevention strategies being neglected, as participants were pre-occupied by the more immediate needs of stroke survivors and their families themselves. Although these findings are specific to the Irish context, they are of international relevance as similar challenges and priorities are likely to be shared across different countries and healthcare systems. Identifying where there are similarities and differences in challenges across healthcare systems is also crucial for identifying potential solutions.

## Conclusion

This study has generated broad and rich evidence on stakeholder views and priorities for improving stroke care in Ireland, as well as a set of specific top priority improvements. Evidence from the literature indicates that many of these concerns are shared across countries. These findings have the potential to inform policy and service development in stroke in Ireland, and also internationally. They also indicate a number of areas for future research. The next step in this project will be to use an already established epidemiological and economic model of stroke [5] to evaluate improvements identified in this research, to provide concrete evidence to policy makers in relation to population demand, cost and cost-effectiveness.

## Supporting information

**S1 File. Consolidated criteria for reporting qualitative studies (COREQ): 32-item checklist.**
(DOCX)

**S1 Table. Survivor interviewee profile (n = 18).**
(DOCX)

**S2 Table. Family/main carer interviewee profile (n = 13).**
(DOCX)

**S3 Table. Senior leader interviewee profile (n = 8).**
(DOCX)

**S4 Table. Phase 1 survey participants profile (professionals) (n = 80).**
(DOCX)

**S5 Table. Phase 2 survey participants profile.**
(DOCX)

**S1 Fig. Identification of priorities by study phase.**
(TIF)

## Acknowledgments

We acknowledge the PPI collaborators who provided critical feedback on the initial research design, and our pilot participants who provided feedback on data collection materials. We also acknowledge the valuable contribution from steering group members Prof Rónán Collins and Dr Maev-Ann Wren. We would also like to acknowledge the substantial support for recruitment from the Irish Heart Foundation support group co-ordinators, the Cork Stroke Support group, and Alzheimer Ireland. Prof Helen Kelly of UCC provided critical advice and support for the development of aphasia-friendly data collection and recruitment materials.

## Author Contributions

**Conceptualization:** Eithne Sexton, Anne Hickey, David J. Williams, Frances Horgan, Elaine Byrne, Chris Macey, Padraic Cuffe, Suzanne Timmons, Kathleen Bennett.

**Formal analysis:** Eithne Sexton, Karen Fowler, Kathleen Bennett.

**Investigation:** Eithne Sexton.

**Methodology:** Eithne Sexton, Karen Fowler, Elaine Byrne, Kathleen Bennett.

**Project administration:** Eithne Sexton.

**Supervision:** Elaine Byrne, Kathleen Bennett.

**Validation:** Anne Hickey, David J. Williams, Frances Horgan, Chris Macey, Padraic Cuffe, Suzanne Timmons, Kathleen Bennett.

**Writing – original draft:** Eithne Sexton.

**Writing – review & editing:** Eithne Sexton, Karen Fowler, Anne Hickey, David J. Williams, Frances Horgan, Elaine Byrne, Chris Macey, Suzanne Timmons, Kathleen Bennett.

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
