## [Decision Letter · Decision Letter 0]

2 Nov 2023

PONE-D-23-30385Priorities for developing stroke care in Ireland from the perspectives of stroke survivors, family carers and professionals involved in stroke care: A mixed methods studyPLOS ONE

Dear Dr. Sexton,

Thank you for submitting your manuscript to PLOS ONE. After careful consideration, we feel that it has merit but does not fully meet PLOS ONE’s publication criteria as it currently stands. Therefore, we invite you to submit a revised version of the manuscript that addresses the points raised during the review process.

Please address in your revision the reviewers' concerns. Key points include enhancing healthcare professional representation, particularly nursing staff, and discussing the impact of stroke severity on patients and caregivers' needs. The exclusion of "atypical presentations" needs better justification or contextualisation, alongside a clearer elaboration on methodology, especially regarding inclusion and exclusion criteria, addressing age bias, and justifying the level of precision. Alos, consider addition of visual aids to clarify research outcomes, deeper exploration of differences between care providers and caregivers/survivors, Discussions on the transferability of findings to settings outside Ireland are also recommended. It is also suggested to condense the methods section for brevity and provide more emphasis on the research gap to improve manuscript clarity and relevance.

We look forward to receiving your revised manuscript.

Kind regards,

Weifeng Han, PhD

Academic Editor

PLOS ONE

Journal Requirements:

Additional Editor Comments:

Please address in your revision the reviewers' concerns. Key points include enhancing healthcare professional representation, particularly nursing staff, and discussing the impact of stroke severity on patients and caregivers' needs. The exclusion of "atypical presentations" needs better justification or contextualisation, alongside a clearer elaboration on methodology, especially regarding inclusion and exclusion criteria, addressing age bias, and justifying the level of precision. Alos, consider addition of visual aids to clarify research outcomes, deeper exploration of differences between care providers and caregivers/survivors, Discussions on the transferability of findings to settings outside Ireland are also recommended. It is also suggested to condense the methods section for brevity and provide more emphasis on the research gap to improve manuscript clarity and relevance.

Reviewers' comments:

Reviewer's Responses to Questions

**Comments to the Author**

1. Is the manuscript technically sound, and do the data support the conclusions?

Reviewer #1: Partly

Reviewer #2: Yes

Reviewer #3: Yes

Reviewer #4: Yes

2. Has the statistical analysis been performed appropriately and rigorously? 

Reviewer #1: N/A

Reviewer #2: Yes

Reviewer #3: Yes

Reviewer #4: N/A

3. Have the authors made all data underlying the findings in their manuscript fully available?

Reviewer #1: Yes

Reviewer #2: Yes

Reviewer #3: No

Reviewer #4: Yes

4. Is the manuscript presented in an intelligible fashion and written in standard English?

Reviewer #1: Yes

Reviewer #2: Yes

Reviewer #3: Yes

Reviewer #4: Yes

5. Review Comments to the Author

Reviewer #1: Introduction section

Past studies have documented the unmet needs of stroke survivors in Ireland, e.g. by Walsh et al. 2015. Please elaborate on how this study will build up the existing knowledge about improvement needed in post-stroke care in Ireland, in relation to the previous studies. In addition, include information about the existing resources and current stroke care system in the introduction. Adding information about the past studies will also help readers to understand the context of this study.

Methodology section

1. Line 129: Remove “at would be”.

2. Participants for both phases: Please elaborate the inclusion and exclusion criteria

3. Majority of the participants (stroke caregivers and stroke survivors) in Phase 1 and Phase 2 were at the older-age group. This may introduce bias in identifying the service priorities. Please explain how you addressed this, and discuss it in the limitation section.

4. Please specify “the reasonable level of precision”, and justification for selecting the precision level.

5. The sample size for survey was small. Please justify how the outcome could represent the service priority at the national level.

6. The percentage for the accepted margin error was inconsistently reported (10% vs 15%). Please explain.

Result section

Include a chart/diagram that shows identification of services priorities from Phase 1 to Phase 2. This will help readers to understand the research process and outcomes better.

Discussion

In addition to rich data, this study considered perspectives from three sources in identifying service priorities. Some differences between the care providers and caregivers/survivors were reported in the survey findings, in Phase 2. Please discuss further.

Reviewer #2: Funding declared.

Ethical approval. Informed consent at all stages.

Would have preferred to see better representation from HCPs inc nursing staff, especially given how registered nurses are the most prevalent professional in the care of individuals with, or at risk of, stroke.

Many thanks for all ancillary material, reviewed and happy with it.

Very glad to see mental health selected by stakeholder meeting as an additional target, as this can be an under-recognised and under-resourced aspect of stroke care – for survivors and family/carers alike.

Overall use of JLA approach appropriate.

Phase 1 – target number of survivors and family/carers met. Happy with targeted approach to recruitment as it is appropriate to ensure capture of data across multiple potential characteristics. Target numbert of survey respondents met.

Clear explanation as to how survey and interview was developed, carried out, and reviewed by interviewees. Very happy to see best practice for cognitive and communication impairments was considered and implemented, as I do feel that those with such impairments can sometimes be under-represented by perceived difficulties in gaining meaningful information. Understandable adjustments w.r.t COVID-19 which I do not believe would have had a meaningful impact on data collection. Interview analysis process explained and appears appropriate.

Phase 2- suitable recruitiment given Phase 1 and approach of whole study, and target met. Again clear how survey was created, adapted and implemented. Interesting that survivors and carers were combined as one group – while it would have been ideal to have sufficient numbers from each to have their own grouping during analysis, I understand the realistic restraints that the team would have been working with. I also acknowledge that this was broken down for the stakeholder meeting. Choice of topics being >20% from either group provides a good approach for narrowing down such a large list of potential topics.

Stakeholder meeting – happy with how this was conducted.

Overall the results from the interviews and surveys align well with previous/recent research, so I am confident the issues identified are relevant and accurate. Acute care is improving across high-income countries, so good to see that reflected in the interviews. I am sadly not surprised that communication is a big barrier in all its guises and forms. Navigating the healthcare system even as working professionals can often take a whole MDT to accurately identify what a person is entitled to and how to access it, so no wonder that it can be a bewildering process for us much less a ‘civilian.’ Maximising therapy is a known barrier to be overcome (see the very recently published NICE guidance, which reinforces more rehabilitation time and a wider considetation of rehabilitation needs!). In particular new and novel approaches to provide targeted neurorehabilitation. Comments from surivors and carers accurately identify the emerging understanding that rehabilitation must encompass cognition, fatigue, and do a much better job of recognising mental ill health. Very interesting to see results for ‘life after stroke,’ as not all studies consider the occupational impact of stroke and stroke recovery, and does align with what is known in this field.

Reviewer #3: The authors did a very thorough subjective assessment of storke survivors, caregivers and HC providers to explore priorities in stroke care. This is a very important, well-done and well written piece that seems relevant also to stroke care outside of Ireland.

Two minor/major comments from me:

(1) First, there is no information on the severity of stroke or the impact the stroke has had on the patients that are interviewed (or their caregivers). Stroke surviovrs with mild disability are likely to have different needs and views than those who survived a severe stroke and may be dependent or even bed bound.

(2) Second, the authors made a unilateral decision to exclude "atypical presentations" because there is "insufficient evidence for effectiveness of potential interventions" and "retention of specialist staff "due to difficulties quantifying a direct benefit for stroke survivors or their families". I agree with leaving those out. However, we need to be aware (readers need to be made aware) that the evidence for "improved access to specialised neuro-rehabilitation" is also very weak and, as mentioned in (1), the need likely differs based on patient need/disability/dependence status.

Reviewer #4: Dear Authors,

This was an interesting read and a worthwhile contribution to the evidence base. I have a few recommendations to improve the article further:

- I would recommend reducing the article, particularly the methods section, in order to increase the likelihood of people reading it.

- The introduction was sound but could have been written in a manner that further emphasised the research gap/necessity for the research. Additionally, giving the reader a clear focus of what the article will discuss in the first paragraph would be useful, as there seemed to be a number of different points put forth.

- Study design - what is meant by "analysed qualitatively" - this is quite vague. I would recommend being more specific.

- Participants - what is meant by "multiple sources" - again, I would recommend being more specific. Additionally, was there any eligibility criteria applied?

- What was the reasoning for thinking that interviews would be easier for survivors and carers, some justification would be useful.

- Incomplete sentence noted on line 129 on page 7.

- Page 10 - line 203 - can you provide a reference for an interpretivist approach?

- Discussion - there were missed opportunities for reference to the wider literature - such as in the third paragraph. Additionally more reference to the wider literature would be useful in all paragraphs under "implications for policy, practice and future research". In the fourth paragraph, you might look to some of the research around mild stroke in relation to "non-traditional stroke symptoms".

- I suggest to remove the sentence on page 37 - "We have explored issues related to navigation..." as I don't believe it adds value.

- Could further discussion of the relevance to other settings outside Ireland be enhanced in order to increase the transferability of findings?

I wish you all the best in the publication process.

6. PLOS authors have the option to publish the peer review history of their article (what does this mean?). If published, this will include your full peer review and any attached files.

Reviewer #1: No

Reviewer #2: No

Reviewer #3: No

Reviewer #4: No

---

## [Author Response · Author response to Decision Letter 0]

20 Dec 2023

Dear Dr Han,

Thank you for the opportunity to revise our paper entitled ‘Priorities for developing stroke care in Ireland from the perspectives of stroke survivors, family carers and professionals involved in stroke care: A mixed methods study’. We are very grateful for the helpful and insightful comments from you and from the reviewers. We have outlined briefly below how each of the concerns highlighted by you have been addressed, followed by detailed response to each specific reviewer comment. We believe that the manuscript has been significantly improved by the requested revisions. We hope that with these added revisions to our paper that it will be deemed suitable for publication in PLOS ONE

Yours Sincerely,

Dr Eithne Sexton and co-authors

Editor comments

Editor comment: Key points include enhancing healthcare professional representation, particularly nursing staff, and discussing the impact of stroke severity on patients and caregivers' needs. 

Response: Thank you for this comment. We agree that participation in the Phase 1 and Phase 2 surveys was somewhat unbalanced in favour of health and social care professionals relative to medical professionals. However, one of the senior leaders we interviewed was a clinical nurse specialist in addition to being a programme manager (and we have added this detail to the Table S3), and one of the professionals at the stakeholder meeting was a clinical nurse specialist. We have included this information at line 582-583. We consider that this ensures that the nurse perspective which we agree is crucial, was included in the process.

Editor comment: The exclusion of "atypical presentations" needs better justification or contextualisation

Response: Thank you for this comment. We have added an additional justification that there is no evidence-based guidelines for improving speed of access for atypical presentations (line 595). See also our response to the Reviewer on page 7 of this Response. 

Editor comment: a clearer elaboration on methodology, especially regarding inclusion and exclusion criteria, addressing age bias, and justifying the level of precision. 

Response: We have specified the inclusion criteria for survivors and carers at line 151-155, for professionals at line 174-176, and we have noted that inclusion criteria for Phase 2 were the same as for Phase 1 (line 191). 

We considered that the level of precision was reasonable given that the survey was part of a broader consensus process, with the results further discussed at a stakeholder meeting. We have added a sentence to clarify this at line 199-201. If we had doubled the total sample size from 96 to 192, this would have improved the margin of error from 10% to 7.5%, which we reasoned would not be a sufficient improvement given the extra time and resources that would have been required to recruit this additional number of participants. 

We do not believe that there is a particular age bias in the study. In Phase 1, close to 50% of participants were aged under 65 (8/18 survivors and 6/13 carers), and in Phase 2 over 50% were aged under 60 (21/33 survivors and 11/17 carers). Considering one quarter of people admitted to hospital in Ireland with stroke are aged under 65, with a median age of 74, we think that on balance the study has included an adequate number of younger survivors (see Irish National Audit of Stroke report, p. 55 https://www.noca.ie/documents/irish-national-audit-of-stroke-national-report-2013-2021). 

Editor comment: Also, consider addition of visual aids to clarify research outcomes

Response: Thank you, this is an excellent idea and addition to the paper. This had been added as S1 Figure to the supplementary material, and referred to in the text in the Results section at line 621-622. 

Editor comment: deeper exploration of differences between care providers and caregivers/survivors, 

Response: Thank you for this helpful comment. We have discussed this in more detail in the Discussion at line 638-642. We note that in Phase 2 more professionals prioritised early supported discharge (ESD) and inpatient rehabilitation. We speculate that this may be related to lower awareness of ESD in the survivor/carer group, and their preference for rehabilitation closer to home, which was apparent from the interviews. 

Editor comment: Discussions on the transferability of findings to settings outside Ireland are also recommended. 

Response: Thank you for this useful comment, we have revised the Implications section in the Discussion to highlight how the findings relate to broader international emerging challenges and policy shifts – for example, the new UK guidelines in relation to increasing intensity of post-stroke rehabilitation (678-80), and the growing interest in the Life After Stroke phase of care, and the current gaps in an established model of care for life after stroke to address this (704-9). 

Editor comment: It is also suggested to condense the methods section for brevity 

Response: Thank you for this very helpful comment, we agree that the Methods section could benefit from being shorter, however we needed to balance this with the need to provide sufficient detail, including requests from reviewers for additional detail, particularly in relation to Participants. We have shortened the Data Collection and Analysis section, removing text at line 247, 256-61, 276 and 322. As we have added significant text to the Discussion in response to the reviewers, we have also made efforts to reduce or remove text – for example at lines 631-5, 666, 709, 769, and 773. Overall, although we have made several additions to the paper in response to the reviews, the word count has only increased by just under 400 words (7674 to 8060).

Editor comment: Provide more emphasis on the research gap to improve manuscript clarity and relevance.

Response: Thank you for these very useful comments. We have revised the Introduction in response to the Reviewer comments, and we think that it is substantially improved as a result. We have included more contextual information about current stroke services in Ireland (line 69-76), referenced the previous study on unmet needs, (line 101-103), and included a clearer statement of how this work addresses a gap and builds on previous work (106-110). We also revised paragraph 1 and 2 to more clearly introduce the focus of the research (line 60-68.). 

Reviewer 1 

Comment: Past studies have documented the unmet needs of stroke survivors in Ireland, e.g. by Walsh et al. 2015. Please elaborate on how this study will build up the existing knowledge about improvement needed in post-stroke care in Ireland, in relation to the previous studies. In addition, include information about the existing resources and current stroke care system in the introduction. Adding information about the past studies will also help readers to understand the context of this study.

Response: Thank you for these very useful comments. We have revised the Introduction in response to your comments, and we think that it is substantially improved as a result. We have included more contextual information about current stroke services in Ireland (line 69-76), referenced the previous study on unmet needs, (line 101-103), and included a clearer statement of how this work addresses a gap and builds on previous work (106-110). We also revised paragraph 1 and 2 to more clearly introduce the focus of the research (line 60-68), in response to comments from Reviewer 4. 

Comment: Line 129: Remove “at would be”.

Response: Thank you very much for spotting this, these words have been removed. 

Comment: Participants for both phases: Please elaborate the inclusion and exclusion criteria

Response: Thank you for this helpful comment. We agree that it would be useful to include more detail in the inclusion criteria. We have specified the inclusion criteria for survivors and carers at line 151-155, for professionals at 175-6, and we have noted that inclusion criteria for Phase 2 were the same as for Phase 1 (line 191). 

Comment. Majority of the participants (stroke caregivers and stroke survivors) in Phase 1 and Phase 2 were at the older-age group. This may introduce bias in identifying the service priorities. Please explain how you addressed this, and discuss it in the limitation section.

Response: Many thanks for this interesting and thought-provoking comment. Perhaps we have misunderstood the comment, but we do not agree that there is a bias against younger age groups in the study. In Phase 1, close to 50% of participants were aged under 65 (8/18 survivors and 6/13 carers), and in Phase 2 over 50% were aged under 60 (21/33 survivors and 11/17 carers). Considering one quarter of people admitted to hospital in Ireland with stroke are aged under 65, with a median age of 74, we think that on balance the study has included an adequate number of younger survivors (see Irish National Audit of Stroke report, p. 55 https://www.noca.ie/documents/irish-national-audit-of-stroke-national-report-2013-2021). 

Comment: Please specify “the reasonable level of precision”, and justification for selecting the precision level. The sample size for survey was small. Please justify how the outcome could represent the service priority at the national level. The percentage for the accepted margin error was inconsistently reported (10% vs 15%). Please explain. 

Response: Thank you for this very helpful comment, we agree that more detail on the sample size and precision is needed. 

We considered that the level of precision was reasonable given that the survey was part of a broader consensus process, with the results further discussed at a stakeholder meeting. We have added a sentence to clarify this at line 199-20§. If we had doubled the total sample size from 96 to 192, this would have improved the margin of error from 10% to 7.5%, which we reasoned would not be a sufficient improvement given the extra time and resources that would have been required to recruit this additional number of participants. 

Apologies for the confusion in relation to the 10% and 15% margin or errors. The 10% margin of error relates to the total sample, while the 15% margin relates to the group-specific estimates. We have revised the text at line 196-199 to make this clear. 

Comment: Include a chart/diagram that shows identification of services priorities from Phase 1 to Phase 2. This will help readers to understand the research process and outcomes better.

Response: Thank you, this is an excellent idea and addition to the paper. This had been added as S1 Figure to the supplementary material, and referred to in the text in the Results section at line 621-2. 

Comment: In addition to rich data, this study considered perspectives from three sources in identifying service priorities. Some differences between the care providers and caregivers/survivors were reported in the survey findings, in Phase 2. Please discuss further.

Response: Thank you for this helpful comment. This has been added to the Discussion at line 638-642. We note that in Phase 2 more professionals prioritised early supported discharge (ESD) and inpatient rehabilitation. We speculate that this may be related to a lower awareness of ESD in the survivor/carer group, and their preference for rehabilitation closer to home, which was apparent from the interviews. 

Reviewer 2 

Comment: 

Comment: Funding declared. Ethical approval. Informed consent at all stages.

Would have preferred to see better representation from HCPs inc nursing staff, especially given how registered nurses are the most prevalent professional in the care of individuals with, or at risk of, stroke.

Many thanks for all ancillary material, reviewed and happy with it.

Very glad to see mental health selected by stakeholder meeting as an additional target, as this can be an under-recognised and under-resourced aspect of stroke care – for survivors and family/carers alike.

Overall use of JLA approach appropriate.

Response: Thank you for the positive feedback, and the helpful comment in relation to HCP representation. We agree that participation in the Phase 1 and Phase 2 surveys was somewhat unbalanced in favour of health and social care professionals relative to medical professionals. However, one of the senior leaders we interviewed was a clinical nurse specialist in addition to being a programme manager (and we have added this detail to the Table S3), and one of the professionals at the stakeholder meeting was a clinical nurse specialist. We have included this information at line 581-582. We consider that this ensures that the nurse perspective which we agree is crucial, was included in the process. 

Comment: Phase 1 – target number of survivors and family/carers met. Happy with targeted approach to recruitment as it is appropriate to ensure capture of data across multiple potential characteristics. Target number of survey respondents met.

Clear explanation as to how survey and interview was developed, carried out, and reviewed by interviewees. Very happy to see best practice for cognitive and communication impairments was considered and implemented, as I do feel that those with such impairments can sometimes be under-represented by perceived difficulties in gaining meaningful information. Understandable adjustments w.r.t COVID-19 which I do not believe would have had a meaningful impact on data collection. Interview analysis process explained and appears appropriate.

Response: Thank you for these positive comments. 

Comment: Phase 2- suitable recruitment given Phase 1 and approach of whole study, and target met. Again clear how survey was created, adapted and implemented. Interesting that survivors and carers were combined as one group – while it would have been ideal to have sufficient numbers from each to have their own grouping during analysis, I understand the realistic restraints that the team would have been working with. I also acknowledge that this was broken down for the stakeholder meeting. Choice of topics being >20% from either group provides a good approach for narrowing down such a large list of potential topics.

Stakeholder meeting – happy with how this was conducted.

Response: Thank you for these positive comments. 

Comment: Overall the results from the interviews and surveys align well with previous/recent research, so I am confident the issues identified are relevant and accurate. Acute care is improving across high-income countries, so good to see that reflected in the interviews. I am sadly not surprised that communication is a big barrier in all its guises and forms. Navigating the healthcare system even as working professionals can often take a whole MDT to accurately identify what a person is entitled to and how to access it, so no wonder that it can be a bewildering process for us much less a ‘civilian.’ Maximising therapy is a known barrier to be overcome (see the very recently published NICE guidance, which reinforces more rehabilitation time and a wider consideration of rehabilitation needs!). In particular new and novel approaches to provide targeted neurorehabilitation. Comments from survivors and carers accurately identify the emerging understanding that rehabilitation must encompass cognition, fatigue, and do a much better job of recognising mental ill health. Very interesting to see results for ‘life after stroke,’ as not all studies consider the occupational impact of stroke and stroke recovery, and does align with what is known in this field.

Response: Thank you for these helpful comments, we agree that the research findings are consistent with previous research, and also highlight emerging areas that have not been considered thoroughly previously, such as the need for long-term supports for Life After Stroke. We agree that the findings are very relevant to the new NICE guidelines on rehabilitation and we have added a reference to these in the Discussion section (line 678-680). 

Reviewer 3

Comment: The authors did a very thorough subjective assessment of stroke survivors, caregivers and HC providers to explore priorities in stroke care. This is a very important, well-done and well written piece that seems relevant also to stroke care outside of Ireland.

Response: Thank you for this positive feedback. 

Comment: First, there is no information on the severity of stroke or the impact the stroke has had on the patients that are interviewed (or their caregivers). Stroke survivors with mild disability are likely to have different needs and views than those who survived a severe stroke and may be dependent or even bed bound.

Response: Thank you for this excellent very useful suggestion. We have added information on the mobility and support needs of the survivors to Tables S1 and S2, and agree this adds important context. 

Comment: Second, the authors made a unilateral decision to exclude "atypical presentations" because there is "insufficient evidence for effectiveness of potential interventions" and "retention of specialist staff "due to difficulties quantifying a direct benefit for stroke survivors or their families". I agree with leaving those out. However, we need to be aware (readers need to be made aware) that the evidence for "improved access to specialised neuro-rehabilitation" is also very weak and, as mentioned in (1), the need likely differs based on patient need/disability/dependence status.

Response: Thank you for these interesting and thought-provoking comments. We have added an additional justification that there is no evidence-based guidelines for improving speed of access for atypical presentations (line 595). We agree that there are limitations to the evidence base in relation to specialised neuro-rehab. However, there is effectiveness evidence based on Cochrane systematic review (see citations at line 604-5), and relevant evidence-based guidelines, including the recent NICE guidance which we have now referenced at line 678-703. We also agree that the need for rehabilitation likely differs across patients, and this is consistent with our finding that rehabilitation must be based on standardised needs-assessment. We have added additional clarification on this at line 680-703. 

Reviewer 4: 

Comment: I would recommend reducing the article, particularly the methods section, in order to increase the likelihood of people reading it.

Response: Thank you for this very helpful comment, we agree that the Methods section could benefit from being shorter, however we needed to balance this with the need to provide sufficient detail, including requests from reviewers for additional detail, particularly in relation to Participants. We have shortened the Data Collection and Analysis section, removing text at line 246, 256-60, 275 and 321. As we have added significant text to the Discussion in response to the reviewers, we have also made efforts to reduce or remove text – for example at lines 631-5, 666, 708, 768, and 772. Overall, although we have made several additions to the paper in response to the reviews, the word count has only increased by just under 400 words (7674 to 8050).

Comment: The introduction was sound but could have been written in a manner that further emphasised the research gap/necessity for the research. Additionally, giving the reader a clear focus of what the article will discuss in the first paragraph would be useful, as there seemed to be a number of different points put forth.

Response: Thank you for these very useful comments. We have revised the Introduction in response to your comments, and we think that it is substantially improved as a result. We revised paragraph 1 and 2 to more clearly introduce the focus of the research (line 60-68) and included a clearer statement of how this work addresses a gap and builds on previous work (106-110). In response to comments from Reviewer 1, we also included more contextual information about current stroke services in Ireland (line 69-76), and referenced previous survey-based research on unmet needs in Ireland (line 101-103). 

Comment Study design - what is meant by "analysed qualitatively" - this is quite vague. I would recommend being more specific. 

Response: Thank you for this very helpful comment, we agree and have revised this sentence to specify that Framework Analysis was used (line 132). 

Comments: Participants - what is meant by "multiple sources" - again, I would recommend being more specific. Additionally, was there any eligibility criteria applied?

Response: Thank you for this helpful comment. We agree that it would be useful to clarify the detail on this. We have added detail on recruitment sources (145-148) We have specified the inclusion criteria for survivors and carers at line 151-155, for professionals at 174-6, and we have noted that inclusion criteria for Phase 2 were the same as for Phase 1 (line 191). 

Comment: What was the reasoning for thinking that interviews would be easier for survivors and carers, some justification would be useful. 

Response: Thank you for this very helpful comment. We have revised this section to clarify our rationale for this (line 180-186). In addition to issues related to computer literacy/technology access, the main issue was that the survey was open-ended requiring potentially lengthy written responses, which we reasoned would have been more effortful and challenging than an interview format. Survivors and carers were given the option of the survey, but more opted for the interview format, consistent with this. 

Comment: Incomplete sentence noted on line 129 on page 7.

Response: Thank you for this comment, this has been corrected. 

Comment: - Page 10 - line 203 - can you provide a reference for an interpretivist approach?

Response: Thank you for this useful comment, a reference has been added at line 246. 

Comment: Discussion - there were missed opportunities for reference to the wider literature - such as in the third paragraph. Additionally more reference to the wider literature would be useful in all paragraphs under "implications for policy, practice and future research". In the fourth paragraph, you might look to some of the research around mild stroke in relation to "non-traditional stroke symptoms".

Response: Thank you for this very useful comment. We have revised the Discussion substantially to include more references to and specific discussion of the wider literature. This includes discussion of the findings in relation to a meta-ethnography of relevant qualitative studies, and two recent studies on survivor/carer priorities (line 643-661). In the Implications section, we have added reference to relevant literature and guidelines on neuro-rehabilitation (678-702), and on interventions and needs relevant to life after stroke (line 705-9). We added a reference to research on the difficulties in speed of access to services for mild stroke at line 746-8. 

Comment: I suggest to remove the sentence on page 37 - "We have explored issues related to navigation..." as I don't believe it adds value.

Response: Thank you for this useful comment. This sentence has been removed. 

Comment: Could further discussion of the relevance to other settings outside Ireland be enhanced in order to increase the transferability of findings?

Response: Thank you for this useful comment, we have revised the Implications section in the Discussion to highlight how the findings relate to broader international emerging challenges and policy shifts – for example, the new UK guidelines in relation to increasing intensity of post-stroke rehabilitation (678-80), and the growing interest in the Life After Stroke phase of care, and the current gaps in an established model of care for to address this (704-9).

---

## [Editor Report · Decision Letter 1]

28 Dec 2023

Priorities for developing stroke care in Ireland from the perspectives of stroke survivors, family carers and professionals involved in stroke care: A mixed methods study

PONE-D-23-30385R1

Dear Dr. Sexton,

We’re pleased to inform you that your manuscript has been judged scientifically suitable for publication and will be formally accepted for publication once it meets all outstanding technical requirements.

Kind regards,

Weifeng Han, PhD

Academic Editor

PLOS ONE

Additional Editor Comments (optional):

Overall, the manuscript is substantially strengthened. This is a rigorous mixed methods study that makes an important contribution by capturing the perspectives of stroke survivors, caregivers and providers on priorities for improving stroke care in Ireland. The findings have relevance both locally and to international efforts to enhance post-stroke services and supports.

The authors have carefully considered all the feedback and made significant revisions to address the issues raised in the review comments. They have provided additional details on the methods, addressed potential biases, justified key decisions, and enhanced the framing and discussion of the findings. The additions of information on stroke severity and mobility/support needs add important context about the participants. The revisions to the introduction and discussion situate the study well in relation to prior research and emerging issues in stroke care and rehabilitation. The authors make a reasonable case that nurse perspectives were adequately captured despite limited nurse participation in the surveys.
---

## [Editor Report · Acceptance letter]

9 Jan 2024

PONE-D-23-30385R1 

PLOS ONE

Dear Dr. Sexton, 

I'm pleased to inform you that your manuscript has been deemed suitable for publication in PLOS ONE. Congratulations! Your manuscript is now being handed over to our production team.

Kind regards, 

on behalf of

Dr. Weifeng Han 

Academic Editor

PLOS ONE